# Investigating Relationships Between Nitrogen Inputs and In-Stream Nitrogen Concentrations and Exports Across Catchments in Victoria, Australia

Olaleye Babatunde[1], Meenakshi Arora[1], Siva Naga Venkat Nara[1], Danlu Guo[2], Ian Cartwright[3], Andrew W. Western[1]

[1]Department of Infrastructure Engineering, University of Melbourne, Australia
[2]School of Engineering, College of Systems & Society, The Australian National University, Canberra, ACT, Australia
[3]School of Earth, Atmosphere and Environment, Monash University, Clayton, Vic, Australia

*Correspondence to:* Olaleye Babatunde (olaleyejohn.babatunde.1@unimelb.edu.au)

**Abstract.** Accurate information on nitrogen (N) inputs to landscapes is crucial for understanding and predicting in-stream N concentrations and downstream N export. However, obtaining detailed catchment-scale data remains challenging due to spatial and temporal variability. We developed a statistical model based on mean annual rainfall to estimate fertiliser N inputs for four agricultural land uses in Victoria, Australia. These estimates, together with contributions from purchased feed, human food N, and biological fixation, were used to (a) examine how N inputs relate to stream total nitrogen (TN) concentrations and export, and (b) assess the influence of rainfall, hydrology, and other catchment characteristics on TN export across 59 diverse catchments. The model revealed a strong positive correlation between average rainfall and fertiliser N input for each land use at the Catchment Management Authority (CMA) (i.e., regional) level, with $R^2$ values ranging from 0.55 to 0.72. Stream TN concentrations were strongly correlated with total N inputs ($R^2 = 0.72$) and fertiliser N inputs ($R^2 = 0.68$). Stream TN export also showed significant relationships with total N inputs ($R^2 = 0.50$) and fertiliser N inputs ($R^2 = 0.53$). The proportion of total N inputs exported varied widely, ranging from 1.4 % to 50 %, with an average of 11 %. This variation was strongly influenced by agricultural activity and hydroclimatic factors. Moreover, the average export proportion was notably lower than values reported for other regions globally, which may reflect Australia's generally lower N input levels. These findings provide a useful tool for water quality assessment and can guide targeted strategies to reduce nitrogen pollution in streams.

## 1 Introduction

Rivers, lakes, and coastal zones around the world are increasingly experiencing rising nitrogen (N) contamination levels, posing serious threats to human well-being and natural habitats (Abascal et al., 2022; Vitousek et al., 1997). While natural factors contribute, human activities play a significant role through point-source discharges and fertiliser application (Angus, 2001; Galloway et al., 2004). The widespread use of synthetic N fertilisers has boosted global food production but has also increased nutrient loads in waterways, causing algal blooms and eutrophication (Fowler et al., 2013). Low nitrogen use efficiency (NUE) from fertiliser application remains a persistent challenge in agricultural landscapes (Stott and Gourley, 2016). For instance, NUE in Australian grazing-based dairy systems declined from 40 % in 1990 to 26 % in 2012 (Stott & Gourley, 2016). In contrast, Europe reported higher NUE levels, averaging 44 % in 2000 and projected to improve to 48 % by 2020 (Oenema et al.,

2009). These inefficiencies indicate that a substantial portion of applied N is not utilized by crops or pastures (Angus and Grace, 2017; Billen et al., 2011). Instead, it may: accumulate in catchments, contributing to delayed N releases into rivers (Van Meter and Basu, 2015); leach into groundwater; be lost via surface runoff; be taken up by vegetation; adsorb onto sediments; or undergo transformation during transport and removal to the atmosphere via biogeochemical processes such as denitrification, anammox, and volatilization (Cellier et al., 2011; Durand et al., 2011; Seitzinger et al., 2006). The dynamics of these processes can vary significantly over time (Guo et al., 2019; Hensley and Cohen, 2020), and are strongly influenced by natural and anthropogenic catchment characteristics (Adame et al., 2021; Deelstra et al., 2014; Helton et al., 2011; Lintern et al., 2018a). Therefore, understanding the factors controlling stream N export is essential for informing effective strategies to reduce nutrient loads to receiving water (Sabo et al., 2019).

Models such as SWAT (Soil and Water Assessment Tool) (Arnold et al., 1998) have been widely used to study N dynamics at various spatial scales within catchments (Gao and Li, 2014; Wellen et al., 2015). However, these process-based models can be challenging to apply because of their extensive input requirements, including climate variables, soil properties, and agricultural management practices such as crop rotations, fertiliser application, and irrigation .In contrast, the Net Anthropogenic Nitrogen Inputs (NANI) method, a mass balance approach, provides a simpler alternative for predicting nutrient loads to receiving waters by quantifying various N inputs on land (Howarth, et al., 1996). Unlike process-based models, which simulate the internal cycling and transformation of nitrogen within the catchment, the NANI approach compares nitrogen inputs to exports without accounting for these internal processes. This methodology has been widely applied in various regions, including North America (Boyer et al., 2002; Goyette et al., 2016; Schaefer et al., 2009), Europe (Howarth et al., 2012), Asia (Zhang et al., 2015), and Africa (Zhou et al., 2014). These studies demonstrate that 70–85 % of the N inputs are retained in the landscape or lost through denitrification (Howarth et al., 2006; Swaney et al., 2012).

Previous studies applying the NANI approach have primarily been conducted in regions characterized by different agricultural practices, typically involving higher fertiliser N application rates compared to those used in Australia (Lu and Tian, 2017a). For instance, wheat, one of Australia's major cereal crops (Australian Bureau of Statistics, 2025), receives considerably less N fertiliser in Australia than in other countries. Ludemann et al. (2022) reported rates of 212 kg ha⁻¹ in China, 78 kg ha⁻¹ in the United States, 85 kg ha⁻¹ in Canada, 150 kg ha⁻¹ in Germany, and 200 kg ha⁻¹ in Chile, compared to just 39 kg ha⁻¹ in Australia. Furthermore, most NANI studies have been conducted in the Northern Hemisphere, primarily in temperate regions with distinct seasonal precipitation patterns. In contrast, Australia's climate is more variable, with milder winters, frequent droughts, and occasional large flood events (Steffen et al., 2018). Additionally, Northern Hemisphere studies typically focus on large catchments (often exceeding 400 km²). Howarth et al. (2012) found that catchments larger than 250 km² exhibit consistent and statistically significant relationships between NANI and stream N flux, whereas smaller catchments show weaker relationships. This raises questions about the applicability of the NANI approach in smaller and more hydrologically variable catchments, such as those in Victoria, Australia where climate and land use vary significantly.

Fertiliser N input, a key component of the NANI framework (Hong et al., 2011), are typically derived from farm-level surveys or disaggregated national statistics. However, these data vary due to inconsistent coverage, differences in reporting periods, and limited information on non-agricultural uses (Sobota et al., 2013). Over time, studies have developed spatial prediction models to estimate fertiliser N use at finer resolutions, using variables

such as fertiliser sales, crop type and acreage, fertiliser expenditure, and climatic data (Stewart et al., 2019; Falcone, 2021). While these approaches can be effective, they often rely on broad assumptions, introducing uncertainty. Moreover, global studies mapping N fertiliser use patterns (Adalibieke et al., 2023; Lu and Tian, 2017b) typically lack the resolution needed to capture fine-scale spatial variability, which is crucial for comparing

N inputs with downstream exports.

Despite global progress in fertiliser N input estimation, region-specific studies are still needed to assess how these inputs translate to stream exports under varied environmental conditions. Most existing research has focused on intensively fertilised systems (Zhang et al., 2015; Howarth et al., 2012), with comparatively little known about N export in landscapes with lower fertiliser use and more variable climatic conditions. In Australia, mass balance

studies, often referred to as nutrient budgets, have been conducted in the subtropical Richmond River catchment (McKEE and Eyre, 2000), Australian dairy farms (Gourley et al., 2007, 2012), dairy farms in Victoria (Rugoho et al., 2018) and two sites in Gippsland, Victoria (Sargent et al., 2025). However, a broader understanding of N input–export relationships across contrasting catchment settings remains limited. This gap reduces the applicability of existing studies for informing management strategies, particularly in regions where N transport is

affected by complex interactions among land use, hydrological flow paths, and biogeochemical processes.

Furthermore, previous Australian studies that developed nitrogen budgets or sought to predict water quality drivers, both of which require accounting for fertiliser inputs, have largely relied on farmer reported data (Gourley et al., 2012), local information from extension officers (Sargent et al., 2025), or region-specific monitoring reports, which often lack precise farm locations (Dairy Australia, 2022). Other studies have used land use percentages as

surrogates for N fertiliser inputs (Guo et al., 2019; Lintern et al., 2018b) or have relied on indirect indicators such as milk production, cattle stocking densities, livestock numbers, or cropped area, rather than directly using fertiliser input data (Lewis et al., 2021; Smith et al., 2013). While these methods have improved our understanding of nutrient dynamics in catchments, they remain limited because land use percentages do not directly correspond to actual N input to the landscape. Moreover, linking N input to export requires data from multiple catchments

(Howarth et al., 2012). Collecting detailed and consistent N inputs from farmers or external agencies is both time-consuming and logistically challenging.

To address this gap, this study developed statistical models to estimate N fertiliser inputs across the four (4) most common agricultural land uses in southeast Australia using average rainfall as the sole predictor. This approach was chosen for two key reasons. (i) In Victoria, most rain-fed agriculture (i.e., dependent on natural

precipitation rather than irrigation) is water-limited, and regions with higher rainfall typically support more intensive agricultural systems. These areas require greater N fertiliser inputs to sustain higher crop and pasture productivity. (ii) Annual rainfall data is readily available and can be easily integrated with land use maps, whereas systematic fertiliser input datasets are lacking. Additionally, we analysed 59 catchments using 11 years of monthly total nitrogen (TN) and streamflow data to examine the relationships between nutrient inputs, in-stream TN

concentrations, and TN export. The influence of hydroclimatic factors on TN export were also examined. The findings aim to enhance our understanding of N fertiliser usage across diverse agricultural landscapes and its links to stream TN export, offering insights to guide sustainability efforts, inform policy, and support adaptive management strategies for improving water quality in Victoria, Australia. The outcomes will also inform research and management strategies in other similar areas globally.


## 2. Method

### 2.1 Catchment characteristics

The catchment boundaries for 137 water-quality monitoring sites were derived from the Geofabric tool (Bureau of Meteorology, 2012). These sites were selected to ensure long-term data availability and broad geographic coverage of diverse land uses and climatic conditions across the state. Water-quality data, including streamflow and TN concentrations, were obtained from the Victorian Water Quality Monitoring Network, which monitors monthly ambient water quality at various sites across the state (see Section 2.8 for details). To focus on rural catchments dominated by agricultural and forested landscapes, catchments with more than 10 % intensive land use were excluded. Intensive land use includes urban areas, industrial zones, infrastructure, commercial developments, mining areas, and other highly modified landscapes, as defined by the Australian Bureau of Agricultural and Resource Economics and Sciences (ABARES, 2016). The final dataset includes 59 rural catchments representing diverse land uses and climatic conditions (Table S1 in the supplementary materials). This selection encompasses a range of agricultural intensities, including some catchments with high forest land use (> 80 %). The inclusion of predominantly forested catchments serves as a reference for understanding natural background N levels, allowing us to distinguish anthropogenic influences from natural N cycling processes across the rural landscape continuum.

Figure 1a presents a land use map of Victoria, showing different land uses and the locations of selected catchments alongside their water quality monitoring sites. Victoria is managed by ten regional Catchment Management Authorities (CMAs), each responsible for water resource management and environmental conservation within its jurisdiction (Fig. 1b). The state's climatic variability is depicted in (Fig. 1c), which illustrates the distribution of average annual rainfall, while (Fig. 1d) represents topographic variation. Elevation (Fig. 1d) is shown for context; it was not used as a predictor due to collinearity with rainfall. The selected catchments range in size from 16 to 11,230 km², with a mean size of 1,284 km² and a median of 458 km², indicating a positively skewed distribution toward larger catchments. Most catchments are located within temperate climatic regions (Peel et al., 2007), where annual average rainfall ranges from 520 mm to 1,800 mm, and temperatures vary between 8 °C and 13 °C. Irrigation is sparse within the study catchments: 0.28 % of land area is classified as irrigated, 28.3 % as non-irrigated, and 71.4 % has no irrigation attribute (Victorian Land Use Information System, 2017). Land use within the catchments is diverse, with forest land use ranging from 1 % to 100 % (Table 1). In this study, catchments with forest land use exceeding 50 % are classified as forested. Conversely, catchments where agricultural land, including cropping, dairy, horticulture, mixed farming and grazing, and non-dairy livestock, accounts for more than 50 % of total land use are classified as agricultural. Agricultural activities dominate across many catchments, particularly mixed farming, which consists of grazing and broad acre rainfed cropping, as well as dairy farming. Intensive uses, cropping, and horticulture make up smaller proportions. The proportions of these land uses, along with other characteristics (including climate, topography, and hydrology), were derived from publicly available datasets (refer to Supplement Table S2 for sources) and were selected due to their well-documented influence on water quality (Lintern et al., 2018a).

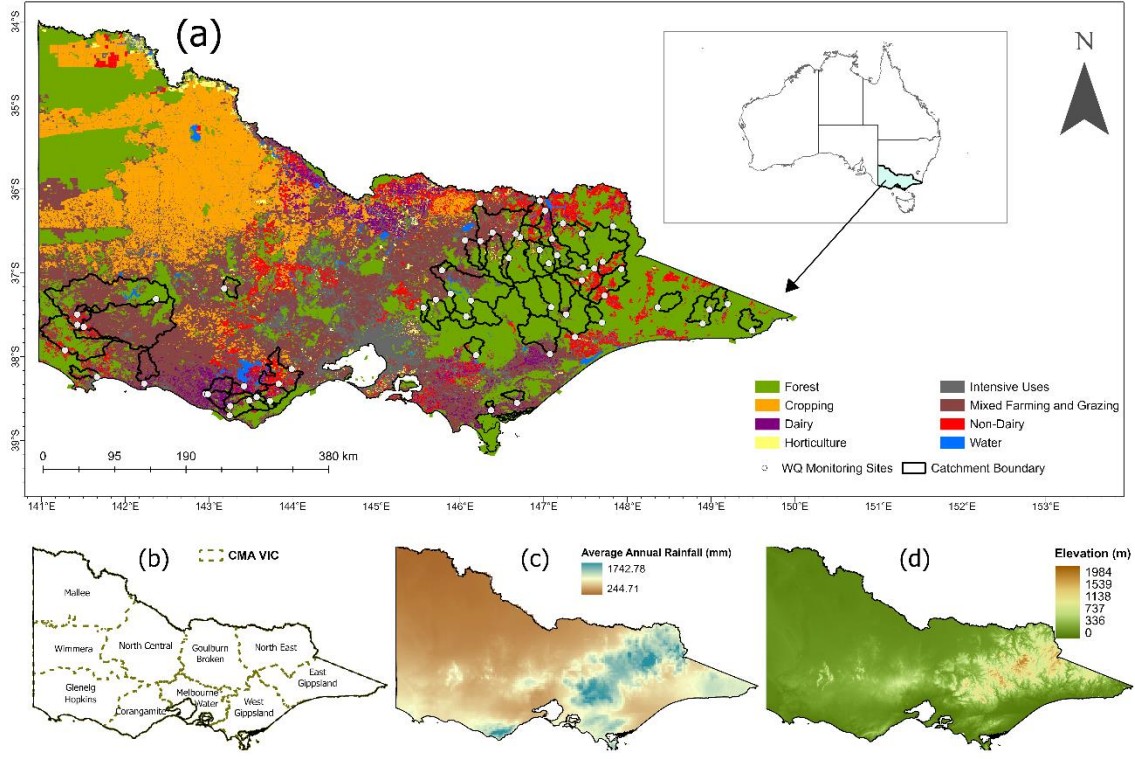

**Figure 1.** Maps showing: (a) Land use across Victoria, including the 59 catchment boundaries and water quality (WQ) monitoring points included in the study. Also inserted is the state of Victoria in Australia (b) Catchment Management Authority (CMA) boundaries. (c) Average annual rainfall distribution (mm). (d) Topography across Victoria (elevation in meters).

**Table 1.** Overview of study catchment characteristics (land use, topography, and climate). Additional detailed data for each catchment are provided in the Supporting Information (Supplement Table S1).

| Catchment Characteristics | Min | Percentile | | | Max | Mean | STD |
|---|---|---|---|---|---|---|---|
| | | 25 % | 50 % | 75 % | | | |
| Land Area Covered by Forests (%) | 1 | 52 | 70 | 94 | 100 | 68 | 28 |
| Land Area Covered by Water (%) | 0 | 0 | 0 | 0 | 2 | 0 | 0 |
| Land Area Used for Intensive Uses (Urban) (%) | 0 | 2 | 4 | 6 | 10 | 4 | 3 |
| Land Area Used for Cropping (%) | 0 | 0 | 0 | 0 | 5 | 0 | 1 |
| Land Area Used for Dairy Farming (%) | 0 | 0 | 0 | 2 | 58 | 5 | 12 |
| Land Area Used for Horticulture (%) | 0 | 0 | 0 | 0 | 5 | 0 | 1 |
| Land Area Used for Mixed Farming and Grazing (%) | 0 | 2 | 5 | 21 | 86 | 16 | 20 |
| Land Area Used for Livestock (Non-Dairy (%)) | 0 | 0 | 2 | 7 | 43 | 6 | 10 |

| | | | | | | | |
|---|---|---|---|---|---|---|---|
| Runoff-Area Normalized Streamflow (mm yr$^{-1}$) | 19 | 100 | 206 | 349 | 1117 | 254 | 210 |
| Annual Temperature (°C) | 8 | 10 | 11 | 12 | 13 | 11 | 1 |
| Catchment Slope (°) | 0 | 4 | 8 | 10 | 19 | 8 | 5 |
| Mean Annual Rainfall (mm yr$^{-1}$) | 675 | 924 | 1093 | 1283 | 1822 | 1111 | 279 |
| Runoff Pereniality (°) | 0 | 2 | 6 | 9 | 17 | 6 | 4 |

**2.2 Nitrogen input data sources**

The primary N input sources considered were fertiliser, purchased feed, human food consumption, and biological fixation. Fertiliser inputs were estimated statistically. Purchased-feed inputs (predominantly in livestock systems) were derived from the Dairy Farm Monitor Project (2016–2017; 2018–2019) and the Livestock Farm Monitor Project (2019–2023). Human food N was calculated from population (WorldPop Australia population-density raster, 1 km, 2020) and national per-capita protein intake (ABS, 2023). Biological N fixation was taken from the

literature (McKee and Eyre, 2000). Supplementary Table S3 summarises the spatial and temporal resolution of all datasets used to estimate N inputs across Victoria, including rainfall (Bureau of Meteorology, 2020), land use (Victorian Land Use Information System, 2017), fertiliser input and irrigation water use (Australian Bureau of Statistics, 2016–2017; 2018–2019), and the dairy and livestock monitor datasets noted above. Average fertiliser input, rainfall, and irrigation water by CMA are reported in Supplementary Tables S4 and S5, and biological

fixation details in Supplementary Table S6.

**2.3 Nitrogen fertiliser estimation**

We estimated the spatial distribution of fertiliser N across four key agricultural land uses in Victoria using mean annual rainfall (MAR) as the predictor. To account for irrigation effects, irrigation depth was added to MAR for

irrigated parcels to derive total water input (TWI). Irrigation was included because it supplements rainfall and influences agricultural intensity. We then estimated fertiliser N as follows:

1.  The MAR raster data was overlaid onto Victoria's land use map using QGIS (QGIS Development Team, 2025). The Zonal Statistics tool was employed to calculate the MAR for all land parcels within the map.

2.  Within CMAs, land parcels were categorized into specific land use types (e.g., Dairy, Cropping, Mixed Farming, Livestock (Non-Dairy)) using the Land Use Description (LU_DESC) field and the Australian Land Use and Management Classification Version 8 (ALUMV8) codes (ABARES, 2016).

3.  For parcels classified as irrigated, the CMA-level mean irrigation depth (ABS 2016–2017; 2018–2019) was added to MAR to derive TWI. This treatment scales fertiliser estimates by TWI and improves model

accuracy in irrigated areas. This adjustment was applied only to the 7.8 % of the mapped area classified as irrigated in the Victorian land-use map. TWI was defined as (Eq 1):

$$TWI_i = \begin{cases} MAR_i + I_i, & \text{if parcel } i \text{ is irrigated} \\ MAR_i, & \text{if parcel } i \text{ is non} - \text{irrigated} \end{cases} \tag{1}$$

where $MAR_i$ is mean annual rainfall for parcel $i$; and $I_i$ is mean irrigation depth (applied only to irrigated parcels).

4. The spatially averaged MAR was then calculated for each land use type within each CMA.

5. A linear regression model was developed to estimate fertiliser N input. The model used the spatially averaged MAR for each land use type within each CMA as the sole predictor, regressed against the fertiliser N input, as reported by the Australian Bureau of Statistics (ABS). This relationship is expressed by Eq. (2):

$$N_{input,i} = \beta_{0,i} + \beta_{1,i} \times MAR_i \tag{2}$$

where $N_{input,i}$ is the estimated fertiliser N input (kg ha$^{-1}$ yr$^{-1}$) for land-use $i$ within a CMA; $\beta_{0,i}$ is the intercept (baseline input at zero MAR); $\beta_{1,i}$ is the coefficient.

6. The regression relationship was used to downscale fertiliser N inputs from the CMA level to individual parcels within each CMA. For irrigated parcels, we substituted TWI (MAR + irrigation; Eq. 1) for MAR; for non-irrigated parcels, the predictor remained MAR.

7. Parcel-level estimates were summed within catchment boundaries to obtain catchment-scale fertiliser inputs (kg ha$^{-1}$ yr$^{-1}$).

**2.4 Disaggregation of nitrogen fertiliser inputs**

For land use-specific regression modelling, as outlined in Section 2.3, ABS-reported fertiliser N inputs were disaggregated. The ABS data provided values for broad categories (e.g., grazing, cropping) but did not distinguish subcategories such as dairy, non-dairy livestock, or mixed farming. Disaggregation was therefore performed using published N application rates from industry reports, as illustrated in the flowchart (Fig. 2). Average N application rates for dairy farms in the Southwest, Northern, and Gippsland regions were sourced from the Dairy Farm Monitor Project (Dairy Australia, 2016–2017; 2018–2019), aligning with the ABS reference years. Rates for non-dairy livestock were derived from the Livestock Farm Monitor Project (Agriculture Victoria, 2020–2023), as data matching the ABS periods were unavailable. A non-dairy-to-dairy ratio was calculated for each region and used to apportion CMA-level grazing N inputs. For example, in northern Catchment Management Authorities (e.g., Northeast, Mallee, North Central), 6% of grazing N inputs were allocated to non-dairy livestock, with the remainder to dairy (Table S4). In mixed farming areas, N inputs were refined using the LU_DESCR_A attribute from the land use map to distinguish between 'Cropping' and 'Grazing-modified Pastures'. Cropping areas were assigned ABS-reported fertilizer N rates, while grazing-modified pastures were assigned the derived non-dairy livestock rates. This process produced land use-specific N input estimates for dairy, non-dairy livestock, and mixed farming areas.

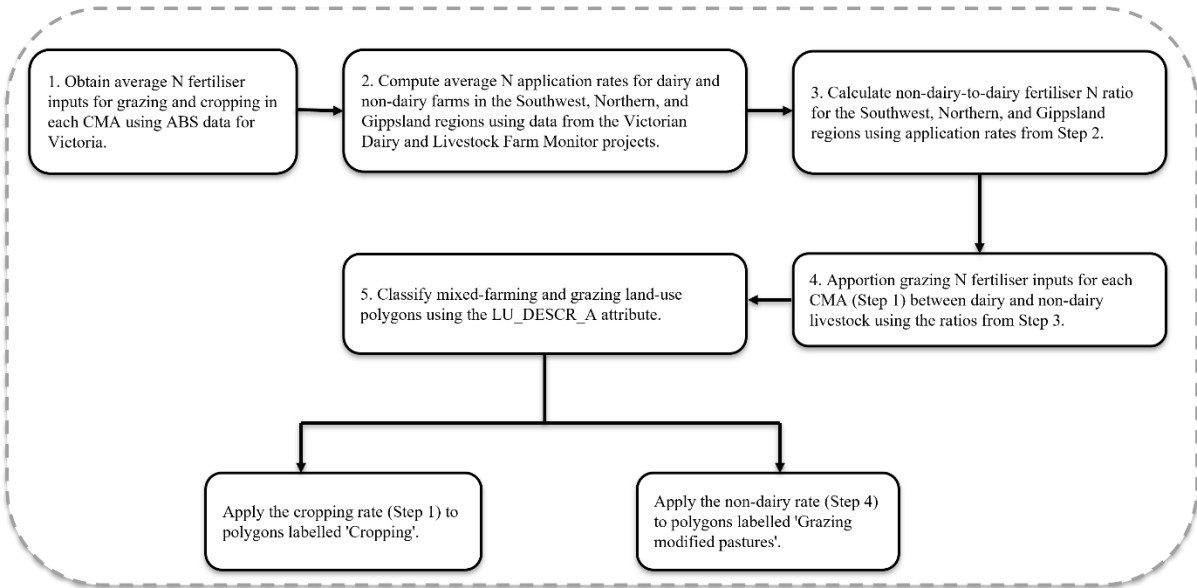

**Figure 2.** Flowchart illustrating the process of disaggregating N fertiliser inputs to dairy, non-dairy livestock, and mixed farming areas using ABS and farm monitor data.

240

## 2.5. Purchased-feed nitrogen inputs

### 2.5.1 Dairy

The N contribution from purchased feed for dairy was estimated from regional stocking rates and purchased-feed quantities. For Gippsland, North Victoria, and Southwest Victoria, we extracted the number of milking cows per usable area (cows ha$^{-1}$) and purchased feed per milking cow (t DM cow$^{-1}$ yr$^{-1}$) from the Dairy Farm Monitor Project for 2016–2017 and 2018–2019 and used the two-year mean for each region (Table S7). These years were selected to align with the fertiliser-N period used in the study. Following Sargent et al. (2025), purchased feed was represented (by dry weight) as 90 % concentrates (2.52 % N), 5% hay (1.71 % N), and 5% silage (2.64 % N), yielding a weighted N fraction of 0.024855 kg N kg$^{-1}$ DM. Per-hectare dairy purchased-feed N rates (kg N ha$^{-1}$ yr$^{-1}$) were computed for each region using Equation S1, then assigned to each catchment according to its region. Catchment-average purchased-feed N was obtained by area-weighting these per-hectare rates by the proportion of dairy land within each catchment.

### 2.5.2 Non-dairy and grazing

For non-dairy and mixed farming systems, purchased-feed N inputs were estimated using regional stocking rates expressed as dry sheep equivalents (DSE ha$^{-1}$) and the proportion of metabolizable energy (ME) supplied by purchased feed, averaged over 2019–2023 (Table S8). A representative purchased-feed composition of 50 % mixed grain, 25 % silage, and 25 % hay (by dry weight) was adopted, giving a weighted N content of 0.0235 kg N kg$^{-1}$ DM and an average ME density of 10.8 MJ kg$^{-1}$ DM (Sargent et al., 2025). An annual ME requirement of 7.6 MJ DSE$^{-1}$ day$^{-1}$ (2,774 MJ DSE$^{-1}$ yr$^{-1}$) was applied to derive dry-matter intake and N contribution per hectare, consistent with standard DSE maintenance benchmarks (OVERSEER Limited, 2018). Regional per-hectare purchased-feed N rates were computed using Eq. S2 and assigned to catchments by region, then area-weighted by the proportion of non-dairy grazing and mixed farming & grazing land within each catchment.

### 2.6 Human nitrogen consumption

Human food N import was estimated as the product of catchment population, national per-capita protein intake (88.3 g person$^{-1}$ d$^{-1}$; ABS, 2023), 365 and 0.16 (protein-to-N conversion; European Food Safety Authority, 2012). Catchment populations were derived from the WorldPop Australia 1-km population-density raster (2020) using zonal statistics in QGIS (QGIS Development Team, 2025); because the raster reports persons km$^{-2}$, totals were calculated as the area-weighted mean density multiplied by mapped catchment area (km²). No adjustment was applied for pre-consumption food waste or retention in the human body.

### 2.7 Other nitrogen inputs

The rate of biological N fixation was not measured in this study. However, the literature reports varying fixation rates for crops, forests, and leguminous plants (Galloway et al., 2004). For example, alfalfa can fix approximately 200 kg ha$^{-1}$ annually, clovers about 150 kg ha$^{-1}$, other forage legumes around 100 kg ha$^{-1}$, and legume–grass pastures approximately 50 kg ha$^{-1}$ (Herridge et al., 2008). McKEE and Eyre (2000) developed nutrient budget studies in the Richmond River catchment, New South Wales, Australia, and extensively reviewed N fixation rates for each agricultural land use. In this study, we used the fixation rate reported in their study for each land use (Supplement Table S6). Point and diffuse sources, such as wastewater treatment plants, urban domestic discharge, and septic systems, were excluded. The study catchments are predominantly agricultural and forested, and in rural Victoria, most treated effluent is irrigated onto land rather than discharged into streams. Moreover, a portion of these pathways is implicitly represented in the human food N input (N imported as food that ultimately contributes to waste and effluent). To avoid double counting, and because intensive urban land occupies <10% of the total area (Section 2.1), we set an explicit wastewater source to zero. Atmospheric deposition was also excluded. Available estimates (e.g., Adams et al., 2014) report only wet deposition and do not distinguish between reduced and oxidized forms or provide the spatial gradients needed for between-catchment comparisons. Due to the inability to reliably constrain its spatial variability with available data, this term was omitted and noted as a limitation.

### 2.8 Stream TN export

Daily streamflow data and monthly measurements of total Kjeldahl nitrogen (TKN) and nitrate/nitrite (NOx) at catchment outlets (Fig. 1a) were obtained for a 23-year period (2001–2023) from the Victorian Water Quality Monitoring Network, accessed through the Water Measurement Information System (Department of Environment, Land, Water and Planning Victoria, 2024). For analysis, an 11-year subset (2013–2023) was used to align with the availability of N fertiliser data (2016–2017 and 2018–2019), allowing for an evaluation of four years before and after the fertiliser data period. TN concentrations in streams were calculated by combining TKN and NOx measurements. Additionally, NOx-to-TN ratios were computed across sites to assess the relative contribution of dissolved nitrate and nitrite to the TN pool and to assess spatial variability in N speciation among catchments.

Areal average TN export (kg ha$^{-1}$ yr$^{-1}$) was estimated using the Weighted Regressions on Time, Discharge, and Season (WRTDS) model (Hirsch et al., 2010), a widely used approach for estimating nutrient exports

(Domagalski et al., 2021; Zhang, 2018). For optimal performance, the WRTDS model is typically applied to datasets spanning at least 20 years, allowing it to capture long-term trends, seasonal variations, and discharge-related fluctuations in water quality (Hirsch et al., 2010). To meet this requirement, the model was run using the full 23-year dataset (2001–2023), generating annual TN exports and concentrations, as well as flow-normalized TN exports and concentrations to assess trends over time. From these results, estimates for the period 2013–2023 were extracted to align with fertiliser input records, as noted earlier. The WRTDS model was implemented in RStudio using the EGRET package (Hirsch et al., 2024) with default argument settings. Briefly, the model was calibrated to data using a 'weighted regression' that assign different weights to individual observations based on their time, discharge and season; the argument settings define the variation of these weights with time, discharge, and season, as well as minimum data required for calibrating the weighted regression. A full description of these parameters is available in Table 7 of the WRTDS user manual (Hirsch and De Cicco, 2015).

The percentage change in flow-normalized TN export over the 11-year period (2013–2023) was computed using the slope from a linear regression of WRTDS output. This percentage change represents the total change over the period relative to the estimated export in 2013. Statistical significance ($p$-values) was determined from the regression analysis of annual flow-normalized TN export values. To assess the role of hydrological variability, flow-weighted TN concentration was calculated as annual TN export divided by annual streamflow. This metric helps distinguish whether variations in TN export are primarily driven by changes in discharge or shifts in nutrient concentrations.

## 3 Results and Discussion

This section presents key findings on N fertiliser inputs across Victoria and N dynamics in the studied catchments. We begin by examining the influence of rainfall on N fertiliser inputs, given its significant role in determining agricultural intensity in Victoria and then assess the spatial distribution of N fertiliser inputs across different land uses (Sections 3.1–3.2). The analysis then shifts to catchment-scale N dynamics, including TN inputs, in-stream TN concentrations, speciation patterns, and TN export rates (Sections 3.3–3.5). Recognizing that high-export catchments pose greater environmental risks, we investigate how land use and hydrological processes influence N speciation and export in these systems (Section 3.6). We then explore the relationship between TN inputs, stream TN concentrations, and export (Sections 3.7–3.8), followed by an identification of key factors influencing percentage of TN exported across catchments (Section 3.9).

### 3.1 Exploring the relationship between mean rainfall and fertiliser N input

Statistically significant positive correlations were found between average rainfall and fertiliser N input across most land use practices ($p < 0.01$), except for dairy, which had a weaker association ($p \approx 0.015$) (Fig. 3). The R² values, ranging from 0.55 to 0.72, indicate that rainfall amounts explained more than half of the variability in N fertiliser use at the CMA level. High rainfall CMAs, such as Glenelg Hopkins, Northeast, Melbourne Water and West Gippsland, consistently receive high fertiliser N input rates across all land uses. This finding aligns with Eckard et al. (2007), who reported N application rates of up to 200 kg ha⁻¹ yr⁻¹ in fertilised dairy pastures in West Gippsland. These zones often use higher fertiliser rates due to intensive agricultural production and are also associated with significant environmental risks, such as nitrate leaching losses (Agricultural Victoria, 2024). In

contrast, the Mallee CMA, characterized by the lowest rainfall in Victoria and a major dryland cropping area, exhibited the lowest N fertiliser inputs across all agricultural land use categories.

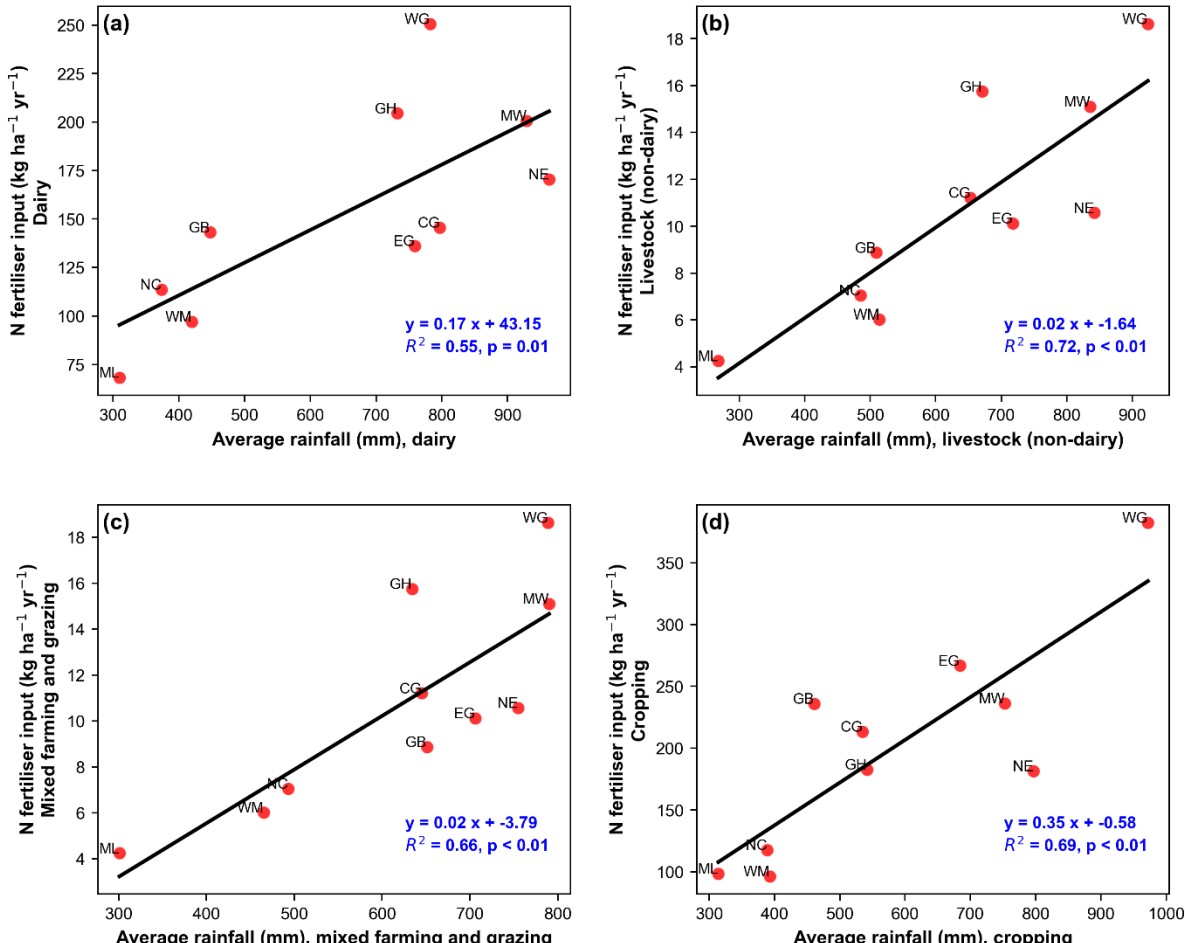

**Figure 3.** Relationships between N fertiliser input and average rainfall across CMAs for: (a) Dairy, (b) Livestock (Non-Dairy), (c) Mixed Farming and Grazing, and (d) Cropping. Abbreviations: ML = Mallee, WM = Wimmera, NC = North Central, GB = Goulburn Broken, CG = Corangamite, GH = Glenelg Hopkins, EG = East Gippsland, NE = Northeast, MW = Melbourne Water, WG = West Gippsland.

The low input in the Mallee is primarily due to the region's semi-arid climate and reliance on rainfall for agriculture, with only 2 % of arable land being irrigated (Mallee Catchment Management Authority, 2020). Due to the limited rainfall and consequent water limitations, agronomic considerations result in lower fertiliser application rates compared to areas with greater water availability. Additionally, dryland crop production in the Mallee often incorporates legumes in crop rotations, where biological N fixation may partially offset the need for synthetic N fertilisers. Furthermore, the highly alkaline soils in the Mallee region (Agricultural Victoria, 2024) may further discourage the application of higher N fertiliser rates.

### 3.2 Spatial distribution of estimated fertiliser N inputs in Victorian agricultural lands

#### 3.2.1 Dairy

As expected, dairy farms received the highest fertiliser N input rates of the four land uses due to their intensive nature, ranging from 150 to 280 kg ha$^{-1}$ yr$^{-1}$ (Fig. 4a). For comparison, Gourley et al. (2012) reported a median fertiliser input of 104.5 kg ha$^{-1}$ yr$^{-1}$ (range: 0–423.9 kg ha$^{-1}$ yr$^{-1}$) based on data from 41 dairy farms across Australia collected during 2008–2009. The 2008–2009 data collection period overlapped with the Millennium Drought, which likely reduced fertiliser application rates and may explain why the median reported by Gourley et al. (2012) is lower than the range observed in this study. Additionally, our estimated fertiliser N input aligns well with the statewide average of 152 ± 95 kg ha$^{-1}$ yr$^{-1}$ (mean ± standard deviation) reported by the Dairy Farm Monitoring Project for the financial years 2016–2017 and 2018–2019 (Dairy Australia, 2016–2019).

Regional variations in fertiliser N input rates on dairy farms in Victoria were evident and primarily influenced by seasonal conditions, soil type, pasture management, and rainfall patterns (Dairy Australia, 2021). Figure 1b shows the map of the CMAs, grouped into broader regions (Gippsland, Southwest, and Northern Victoria) based on their geographical distribution and alignment with the Dairy Farm Monitor Project (DFMP; Dairy Australia, 2021) classification, which accounts for regional differences in climate, land use, and dairy production. Dairy Australia (2017, 2019) reported average N input rates of 193 ± 105 kg ha$^{-1}$ yr$^{-1}$ (n = 50) in Gippsland, 149 ± 86 kg ha$^{-1}$ yr$^{-1}$ (n = 50) in the Southwest, and 115 ± 77 kg ha$^{-1}$ yr$^{-1}$ (n = 50) in the Northern region, where n refers to the number of farms or land parcels. In comparison, our analysis estimated higher N application rates across all regions: 212.01 ± 31 kg ha$^{-1}$ yr$^{-1}$ (n = 4,236) in Gippsland, 196.60 ± 32 kg ha$^{-1}$ yr$^{-1}$ (n = 8,085) in the Southwest, and 181.27 ± 37 kg ha$^{-1}$ yr$^{-1}$ (n = 10,558) in the Northern region. Supplement Fig. S1 illustrates trends in N fertiliser usage across regions. The higher rates in Gippsland can be attributed to intensively grazed pastures associated with dairy and beef cattle, combined with high rainfall and irrigation in the region.

#### 3.2.2 Non-dairy

Livestock farms, including sheep, beef, and lamb, exhibited low N inputs, with values below 30 kg ha$^{-1}$ yr$^{-1}$ (Fig. 4b). This finding aligns with the statewide farm-level average of 13 ± 22 kg ha$^{-1}$ yr$^{-1}$, reported for these systems for the years 2019–2020 and 2020–2021 (Supplement Fig. S2; Agriculture Victoria, 2022). Unlike dairy operations, which require high N inputs to sustain intensive pasture production, non-dairy livestock farms typically have lower nutrient demands. These farms often rely on alternative N sources, such as native soil N and biological N fixation by leguminous crops, to meet their N requirements (Angus, 2001). Regional averages reported by the Livestock Monitoring Project for 2019–2020 and 2020–2021 indicated slightly higher N inputs in the Southwest: 16.32 ± 25 kg ha$^{-1}$ yr$^{-1}$ (n = 102), compared to Gippsland: 14.02 ± 23 kg ha$^{-1}$ yr$^{-1}$ (n = 44), and the North: 7.91 ± 15 kg ha$^{-1}$ yr$^{-1}$ (n = 61). Our analysis revealed similar but slightly higher regional averages: 18.55 ± 4 kg ha$^{-1}$ yr$^{-1}$ (n = 6,901) in Gippsland, 16.03 ± 4 kg ha$^{-1}$ yr$^{-1}$ (n = 9,901) in the South, and 14.25 ± 7 kg ha$^{-1}$ yr$^{-1}$ (n = 26,225) in the North. The estimated N fertiliser input values for non-dairy livestock farms in Victoria are consistent with the average of 10 kg ha$^{-1}$ reported for New Zealand sheep and beef farms (Parfitt et al., 2012). This similarity suggests that low N input rates in these systems are a common practice not only in Victoria but also in other regions with comparable agricultural systems and climatic conditions.

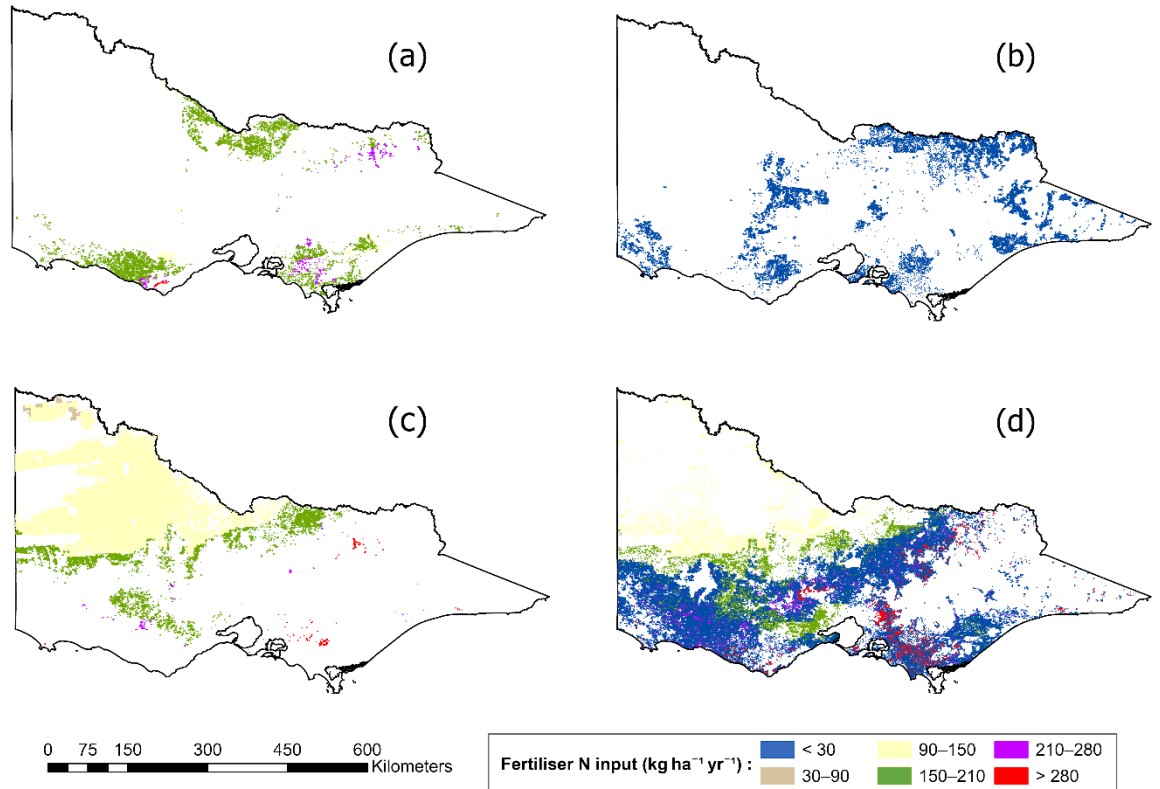

**Figure 4.** Spatial distribution of estimated fertiliser nitrogen input by land use in Victoria: (a) Dairy, (b) Non-dairy, (c) Cropping, and (d) Mixed farming and grazing.

### 3.2.3 Cropping

Most crop production in Victoria consists of broadacre rainfed cropping, primarily of cereals like wheat, concentrated in the northern regions (ABS, 2018). In our dataset, fertiliser N inputs were reported under a single "cropping" category, combining broadacre and horticultural land uses due to the unavailability of separate horticultural data. While horticultural systems typically employ more intensive nutrient management (e.g., higher fertiliser rates) than broadacre cropping, we present combined fertiliser N input estimates for this category due to data limitations and the small fraction of cropping area dedicated to horticulture. The estimated average fertiliser N input for cropping, encompassing horticulture, irrigated cereals, and dryland crops (e.g., wheat, barley, canola, sorghum, oats, and triticale), ranges from 90 to 210 kg ha$^{-1}$ yr$^{-1}$ (Fig. 4c). These estimates, which vary by crop type and management practices (Agriculture Victoria, 2022), align with reported Australian averages. Angus and Grace (2017), noted fertiliser N inputs of 300 kg ha$^{-1}$ yr$^{-1}$ for cotton, 100 kg ha$^{-1}$ yr$^{-1}$ for horticulture and irrigated cereals, and 45 kg ha$^{-1}$ yr$^{-1}$ for dryland crops. Similarly, Agriculture Victoria (2022) reported an average farm-level N input of 60 kg ha$^{-1}$ yr$^{-1}$ for wheat, barley, canola, and lupins in Victoria, supporting the validity of our findings.

### 3.2.4 Mixed farming and grazing

Mixed farming areas exhibit considerable variability in N inputs, particularly in zones where both cropping and livestock (non-dairy) operations are present. Cropping areas in mixed farming systems display a wide range of N

inputs, averaging between 90 and 150 kg ha⁻¹ yr⁻¹ (Fig. 4d). This variability is likely influenced by differences in crop types and the agricultural practices employed in these regions. In contrast, the non-dairy livestock component, which accounts for approximately 80 % of mixed farming areas, received significantly lower N inputs, with values below 30 kg ha⁻¹ yr⁻¹.

### 3.3 Catchment nitrogen inputs

The data from Fig. 4 was aggregated to calculate the average fertiliser TN input across each catchment, then contributions from purchased feed, human food N, and biological fixation were added. Total inputs ranged from 5–184 kg ha⁻¹ yr⁻¹ (mean 32 kg ha⁻¹ yr⁻¹), and the mean fertiliser-N input was 16 kg ha⁻¹ yr⁻¹ (Supplementary Table S9). Most catchments (≈80%) fell between 5 and 25 kg ha⁻¹ yr⁻¹ (Fig. 5a), with a small number exhibiting substantially higher values. The highest inputs (101–188 kg ha⁻¹ yr⁻¹) occurred in southern Victoria and were predominantly associated with intensive dairy agriculture. N fertiliser was identified as the largest individual input source and dominated in catchments with higher input rates. Positive correlations were observed between Total N inputs and the percentage of agricultural land use ($R^2 = 0.57$, $p < 0.001$) and dairy land area ($R^2 = 0.91$, $p < 0.001$). In contrast, forest land use showed a negative correlation with N inputs ($R^2 = 0.58$, $p < 0.001$; Supplement Fig. S3), which is expected since N input into forested areas in this study area is low and primarily depends on biological N fixation by legumes (including acacia species) and atmospheric deposition. The average N fertiliser input in our study is substantially lower than fertiliser inputs reported for intensively farmed catchments such as the Huai River Basin, China (188 kg ha⁻¹ yr⁻¹; Zhang et al., 2015) and the Yangtze River Basin (18–42 kg ha⁻¹ yr⁻¹; Chen et al., 2016).

Purchased-feed N (dairy and non-dairy grazing) ranged from 0 to 34.6 kg ha⁻¹ yr⁻¹ across catchments. Dairy accounted for most of this input, with a mean of 3.2 kg N ha⁻¹ yr⁻¹, while non-dairy contributed a mean of 0.47 kg N ha⁻¹ yr⁻¹. Human food N was comparatively small, ranging from 0 to 0.16 kg ha⁻¹ yr⁻¹. Biological N fixation estimates (5–29 kg ha⁻¹ yr⁻¹) were comparable to the median value of 13 kg ha⁻¹ yr⁻¹ reported for Australian dairy farms (Gourley et al., 2012) and similar to those estimated for California watersheds (Sobota et al., 2009). However, these values were lower than those reported for New Zealand ( Ledgard et al., 1999; Parfitt et al., 2012). Biological N fixation remains an area with little available data.

### 3.4 TN concentration and speciation patterns

Our analysis revealed considerable variation in mean TN concentrations of surface water across the study catchments. TN concentrations ranged from 0.11 to 2.41 mg L⁻¹ (Fig. 5b), with an average of 0.63 mg L⁻¹. The highest mean TN concentration (2.41 mg L⁻¹) was observed at Scotts Creek at Curdie and Pirron Yallock Creek (Supplement Table S9), where agriculture (mainly dairy) accounts for 84.2 % and 78.8 % of the catchment area, respectively. This aligns with statewide TN estimates from the Victorian Water Quality Analysis (2022), which report that catchments dominated by agricultural land use tend to have the highest TN concentrations, ranging from 1.2 to 2.6 mg L⁻¹. Our findings are consistent with  Liu et al. (2022), who reported mean TN concentrations of 0.6 mg L⁻¹ in temperate Australian streams. However, the mean TN concentration in our study (0.63 mg L⁻¹) is less than half of the U.S. average of 1.37 mg L⁻¹ reported by Bellmore et al. (2018) for streams across the conterminous United States (range: 0.015–34.45 mg L⁻¹). This discrepancy likely reflects differences in nitrogen

input intensity, land use composition, and hydrological processes between the two regions, all of which influence stream nitrogen concentrations (Lintern et al., 2018a).

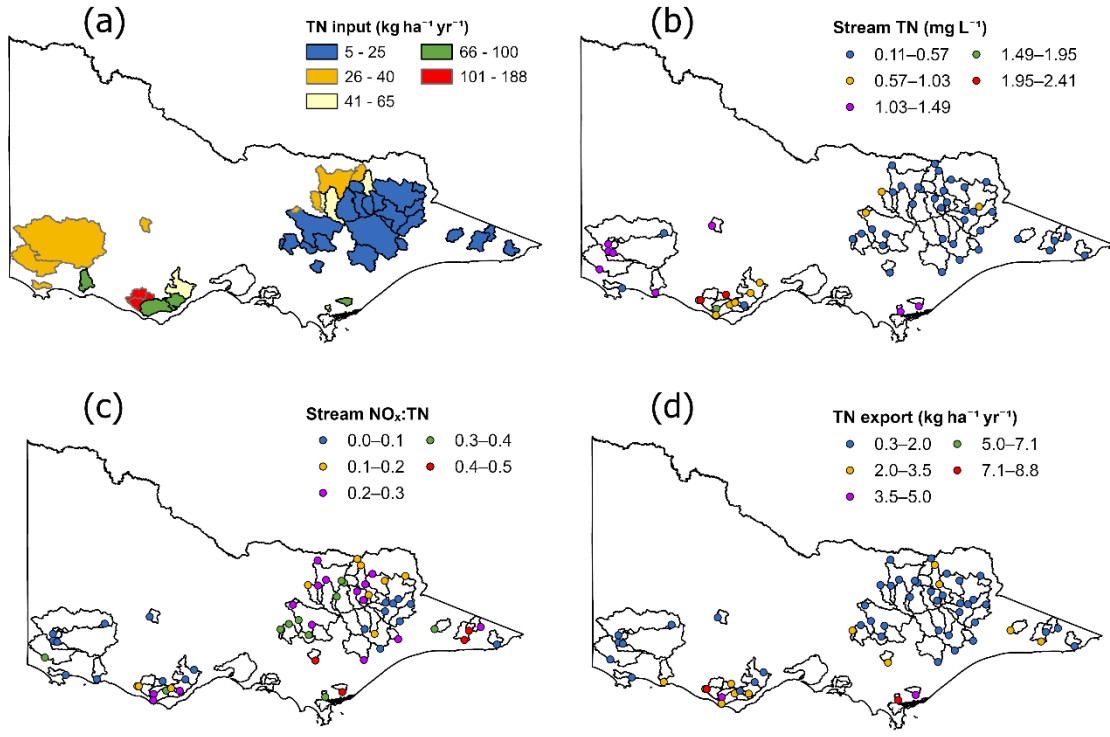

**Figure 5.** Map showing (a) TN input, (b) Stream TN concentration, (c) Stream NOx: TN ratio, and (d) TN export. Data points represent individual catchments.

Nitrogen speciation patterns across all catchments indicate that TN is primarily composed of total Kjeldahl nitrogen (TKN) (Supplement Table S9), though the relative proportions of NOx-N and TKN vary between sites (Fig. 5c). The highest NOx: TN ratio (0.5) was observed in forest-dominated catchments, while lower ratios occurred in both forested and agricultural catchments. These findings suggest that N speciation is not exclusively linked to a specific land use type; rather, other factors such as hydrological conditions, N cycling processes, and catchment characteristics likely influence the observed patterns.

**3.5 TN export**

TN export from the catchments ranged from 0.3 to 8.8 kg ha⁻¹ yr⁻¹, with an average value of 1.7 kg ha⁻¹ yr⁻¹ across all catchments (Supplement Table S9; Fig. 5d). This variation is expected, as TN export is influenced by factors such as climate, hydrology, soil type, and land use (Drewry et al., 2006). The average TN export observed in this study is lower than the land use–specific values reported for Australian field studies by Drewry et al. (2006), which estimated approximately 13 kg ha⁻¹ yr⁻¹ for dairy, 3 kg ha⁻¹ yr⁻¹ for sheep and cattle, and 1 kg ha⁻¹ yr⁻¹ for forest. This lower value likely reflects the higher proportion of forest in our study catchments, as forested areas generally contribute less N export. The observed average is also lower than broader regional estimates, with N

export in the Saint Lawrence Basin, Canada, ranging between 0 and 18 kg ha$^{-1}$ yr$^{-1}$ (Goyette et al., 2016), and from 0 to 20 kg ha$^{-1}$ yr$^{-1}$ across U.S. catchments (Boyer et al., 2002; Schaefer and Alber, 2007).

**3.6 High export catchments**

Trends in TN export for high-export catchments (> 2.5 kg ha$^{-1}$ yr$^{-1}$) are shown in Fig. 6. These catchments present significant environmental risks due to their elevated nutrient contributions, which can increase downstream nitrogen loads and disrupt nutrient balances in receiving waters. Understanding these catchments is crucial for identifying how hydrological processes influence their export patterns. For most sites, the flow normalized trends in TN export were relatively small, with percentage changes ranging from 2.2 % to 198 % over the eleven-year

period. These trends were not statistically significant ($p > 0.05$) at sites such as Gellibrand River at Bunker (2.2 %), Barwon River East Branch at Forrest (39.8 %), and Tarra River at Yarram (51.0 %). However, four catchments, Curdies River at Curdie (197.8 %), Pirron Yallock at Pirron Yallock (161.1 %), Scotts Creek at Curdie (147.9 %), and Kennedys Creek at Kennedy (130.6 %) exhibited statistically significant increases in TN export ($p \leq 0.05$) (Supplement Table S10; Fig. S4).

Annual exports show substantial variability across many catchments, primarily driven by flow changes. This relationship is evident from the flow-weighted concentration trends (Supplement Fig. S5) and the coefficients of variation (CV) of flow and flux (Supplement Fig. S6). In Scotts Creek and Kennedys Creek, the CV values for flow (43.4 % and 38.9 %) closely match those for flux (43.5 % and 36.9 %), indicating that flow is the dominant driver of export variability in these catchments. While there was a significant increase in TN export over the study

period, most of the change in export in these catchments is related to changes in discharge. However, at some sites, such as the Tarra River at Yarram and the Tanjil River at Tanjil Junction, the variability in nitrogen flux far exceeds that in flow. The CV values for nitrogen flux and flow are 108.8 % and 72.7 % in the Tarra River, and 69.5 % and 33.7 % in the Tanjil River, suggesting additional factors influence TN export variability. Furthermore, a closer examination of a few catchments within the high TN export sites (> 2.5 kg ha$^{-1}$ yr$^{-1}$) reveals that Agnes

River (AR) has a NOx: TN ratio of 0.4, whereas Scott's Creek (SC) and Kennedy Creek (KC) have lower ratios of 0.2 and 0.3, respectively (Supplement Table S9). The elevated NO$_x$: TN ratio observed in AR suggests a greater contribution from deeper groundwater flow pathways rather than surface runoff (Smith et al., 2013). These pathways are associated with longer transit times (Cartwright et al., 2020), which facilitate the mineralization and nitrification of retained organic nitrogen and ammonium (NH$_4^+$) into nitrate (NO$_3^-$). However, while these transit

times are sufficiently long to enhance nitrification, they are not prolonged enough to enable substantial denitrification. As nitrate is highly mobile and less susceptible to attenuation, it persists in these deeper pathways, contributing to the elevated NO$_x$: TN ratio without significant reduction to gaseous nitrogen forms through denitrification.

This pattern is further supported by correlation analysis, which shows a significant positive relationship

between forest land use and the NOx: TN ratio in high TN export sites (> 2.5 kg ha$^{-1}$ yr$^{-1}$) (Supplement Table S11: r = 0.680, $p = 0.032$). With 52 % of AR's land covered by forests (Supplement Fig. S7), this finding strongly suggests that forested land contributes to the higher proportion of nitrate in AR's streamflow. Notably, this positive relationship between forest land use and the NOx: TN ratio is also evident in low-export sites (< 2.5 kg ha$^{-1}$ yr$^{-1}$; Table S11: r = 0.425, $p = 0.002$), indicating that more forested catchments like AR have a consistently

higher proportion of nitrogen export as nitrate across different N export levels. When considering all catchments

combined, forest land use remains positively correlated with the NOx: TN ratio (Supplement Table S11: r = 0.32, *p* = 0.01), though the correlation is weaker than in high-export catchments. This consistency highlights the important role of forest land use in nitrate export, particularly in AR, where subsurface flow pathways deliver nitrate to streamflow even without N fertiliser inputs.

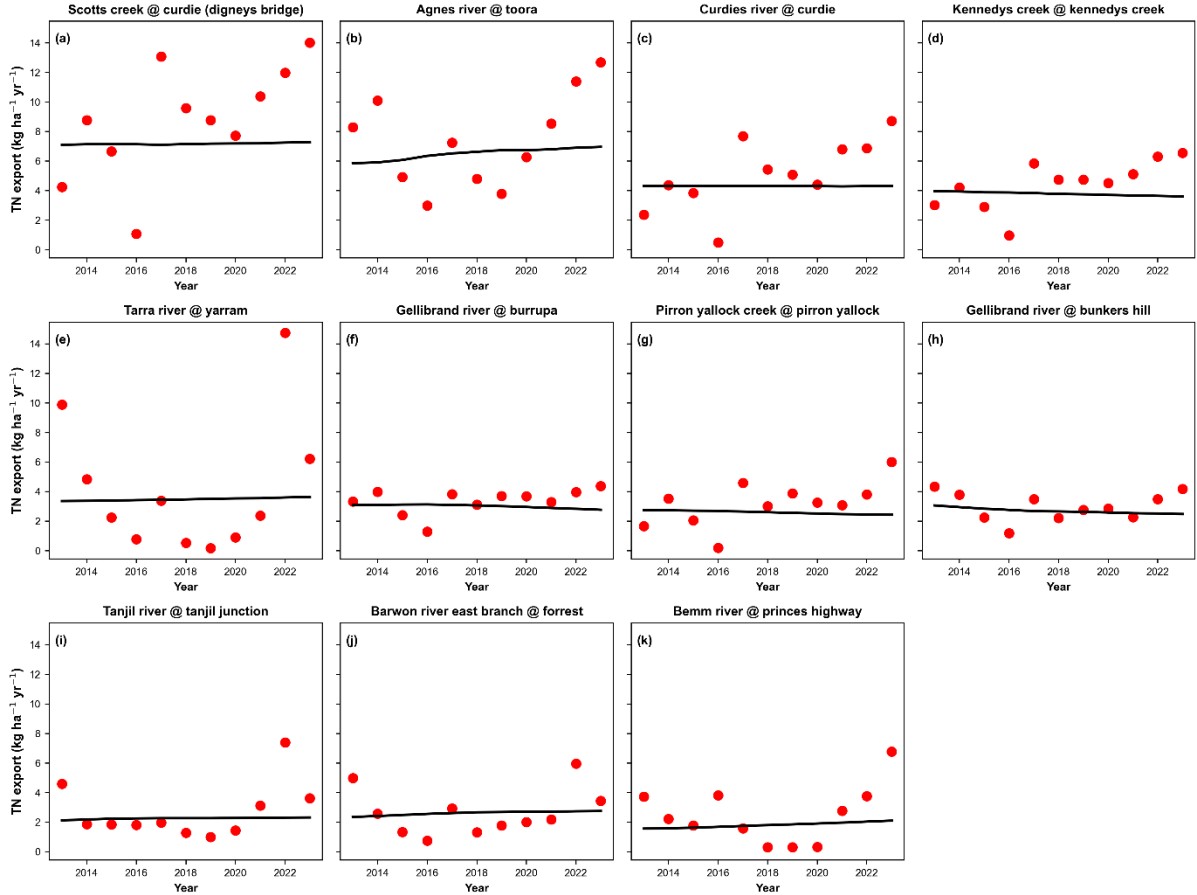

**Figure 6.** TN export at high-export sites. Annual mean TN exports are shown as red dots, while flow-normalized TN exports, representing long-term trends, are shown as a black line, estimated using the WRTDS model.

In contrast, SC and KC exhibit lower NOx: TN ratios, likely due to the greater proportion of agricultural land use in these catchments (Supplement Fig. S7). This pattern is consistent with findings by Adelana et al. (2020), who observed that N contributions from synthetic fertilisers are often lower than those from the wash-off of animal waste in dairy farms in West Gippsland, Victoria. The significant negative correlation between dairy land and the NOx: TN ratio in high TN export sites (Supplement Table S11: r = −0.730, *p* = 0.016) suggests that streams in these catchments contain a higher proportion of organic nitrogen or ammonia, driven by intensive dairy farming activities and associated animal waste runoff (Drewry et al., 2006; Rawnsley et al., 2019). Additionally, horticulture, livestock (non-dairy), and mixed farming and grazing in the high-export sites exhibited weaker correlations with the NOx: TN ratio. This is likely due to the diverse N sources and variable management practices associated with these land uses, including synthetic fertilisers, irrigation methods, and crop rotation strategies. These factors may dilute or obscure consistent patterns in N speciation. The absence of significant relationships highlights the dominant roles of forests and dairy farming in influencing nitrogen speciation in these high export sites.

**3.7 Stream TN and its relationship with inputs**

TN concentrations showed a significant positive correlation with total N inputs, explaining 72 % of the variance ($R^2 = 0.72$, $p < 0.001$; Fig. 7a). This finding aligns with previous studies, such as Sobota et al. (2009), who reported a correlation of $R^2 = 0.41$ in California watersheds, and Jiajia Lin et al. (2021) and Bellmore et al. (2018), who observed R² values of 0.57 and 0.42, respectively, across the conterminous United States. N fertiliser input alone also exhibited a significant correlation with stream TN concentration ($R^2 = 0.68$, $p \leq 0.001$; Fig. 7b). Positive correlations were also observed between stream TN concentrations and agricultural land use percentage (Fig. 7c; $R^2 = 0.68$, $p \leq 0.001$), dairy land area ($R^2 = 0.61$, $p \leq 0.001$; Fig. 7d), and mixed farming and grazing land area percentage ($R^2 = 0.33$, $p \leq 0.001$; Fig. 7e). Conversely, forest land use exhibited significant variability and a negative correlation with stream TN concentrations ($R^2 = 0.67$, $p \leq 0.001$; Fig. 7f). This may be attributed to both lower anthropogenic N inputs in forested catchments and natural retention mechanisms such as vegetation uptake and reduced surface runoff. This aligns with findings by Djodjic et al. (2021), who demonstrated that increased forest and wetland coverage reduces nutrient concentrations in streams.

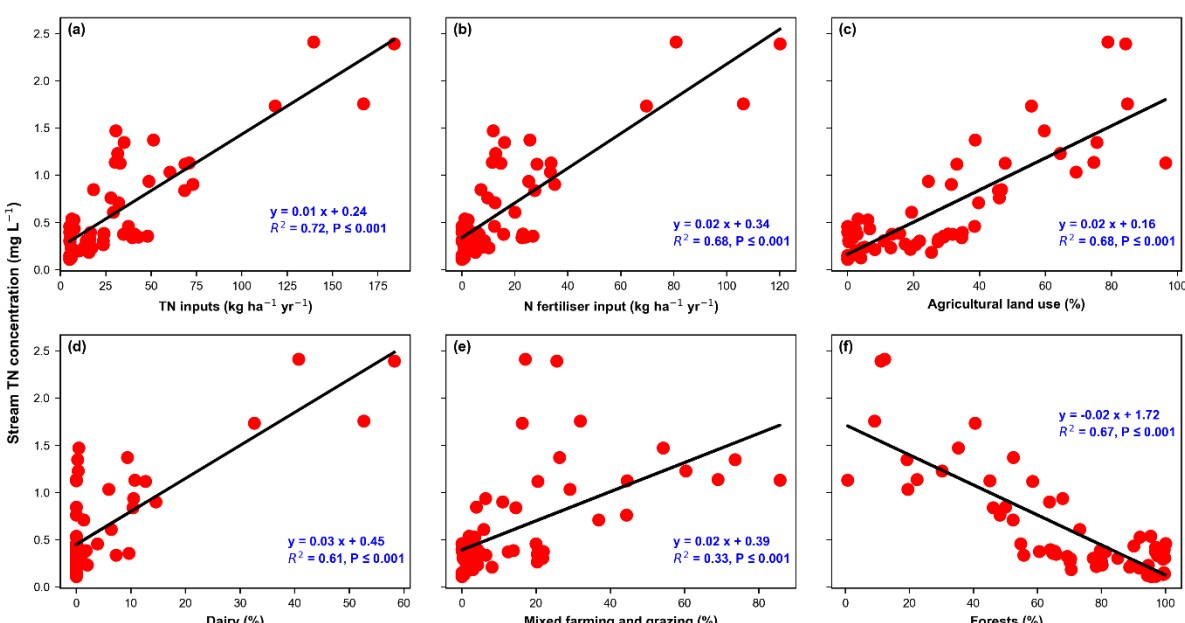

**Figure 7.** Correlation of stream TN concentration with (a) TN input, (b) N fertiliser input, (c) Agricultural land use, (d) Dairy, (e) Mixed farming and grazing, and (f) Forests.

**3.8 Relating inputs to stream TN export**

There is a positive correlation between stream TN export and total N inputs across the catchments (Fig. 8a; $R^2 = 0.50$, $p < 0.001$). This finding aligns with studies conducted in various regions, which have reported R² values ranging from 0.48 to 0.87. The strength of this relationship may vary due to regional differences in hydrology, land use, nitrogen retention, and climate. For example, McKEE and Eyre (2000) reported a similar R² of 0.47 in the Richmond River catchment, Australia, likely due to comparable total N input ranges (12–57 kg ha⁻¹) relative to the mean value observed in this study (32 kg ha⁻¹). In contrast, Goyette et al. (2016) reported a stronger correlation ($R^2 = 0.87$) between N inputs and stream export across catchments in the St. Lawrence Basin, where

inputs ranged from 1.1 to 93.5 kg ha⁻¹ yr⁻¹. They noted that this relationship was largely driven by catchments with high N inputs, which exert a strong influence on the slope of the regression. Similarly, Schaefer and Alber. (2007) observed high correlations ($R^2 = 0.80$) in northern and mid-Atlantic U.S. watersheds but a weaker relationship ($R^2 = 0.53$) in southeastern U.S. watersheds. European regional watersheds also showed moderate correlations, with an $R^2$ of 0.54 (Billen et al., 2011).

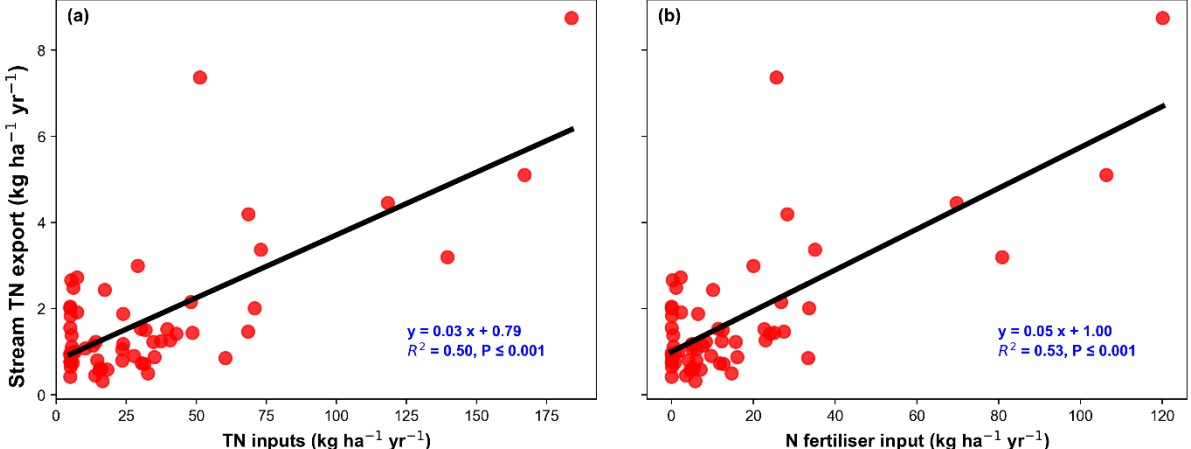

**Figure 8**. Correlation of stream TN export with (a) TN inputs, (b) N Fertiliser input

The slope of the regression between total N inputs and stream TN export (Fig. 8a) across all catchments is 0.03 ($p < 0.001$), indicating that only 3 % of total N inputs were exported as stream fluxes. However, the export-to-input ratio varies considerably among individual catchments, ranging from 1.4 to 50 % (Supplement Table S9; see also Supplement Table S1 for full catchment-level data). This variability is particularly pronounced in low-input catchments, where biological N fixation constitute a greater share of the total input. This variability is particularly pronounced in low-input catchments, where biological nitrogen (N) fixation constitutes a greater share of the total input. This source is less well characterized, was estimated more simply, and thus has greater uncertainty, which may partially explain these differences. A significant correlation was also observed between fertiliser N input and stream TN exports (Fig. 8b; $R^2 = 0.53$, $p < 0.001$).This confirms that catchments with fertiliser application to a greater fraction of their area export more N to streams, a pattern consistent with global studies that identify synthetic fertilisers as a major driver of N export in catchments (Howarth et al., 2012; Schaefer et al., 2009).

### 3.9 Factors influencing export as a percentage of TN input

Our analysis indicates that runoff (representing area-normalized streamflow; Fig. 9a; $R^2 = 0.22$, $p < 0.001$) and precipitation (Fig. 9b; $R^2 = 0.18$, $p < 0.001$) are strong predictors of percentage of TN export across the catchments. Among the catchment characteristics examined, runoff perenniality defined as the proportion of time a stream has continuous flow throughout the year, emerged as the strongest driver (Fig. 9c; $R^2 = 0.29$, $p < 0.001$). Temperature showed a weak negative correlation with TN export percentage (Fig 9d; $R^2 = 0.06$, $p = 0.058$), while catchment slope did not exhibit a significant correlation (Fig 9e; $R^2 = 0.04$, $p = 0.130$). Stream density exhibited a moderate positive correlation (Fig 9f; $R^2 = 0.12$, $p = 0.008$).

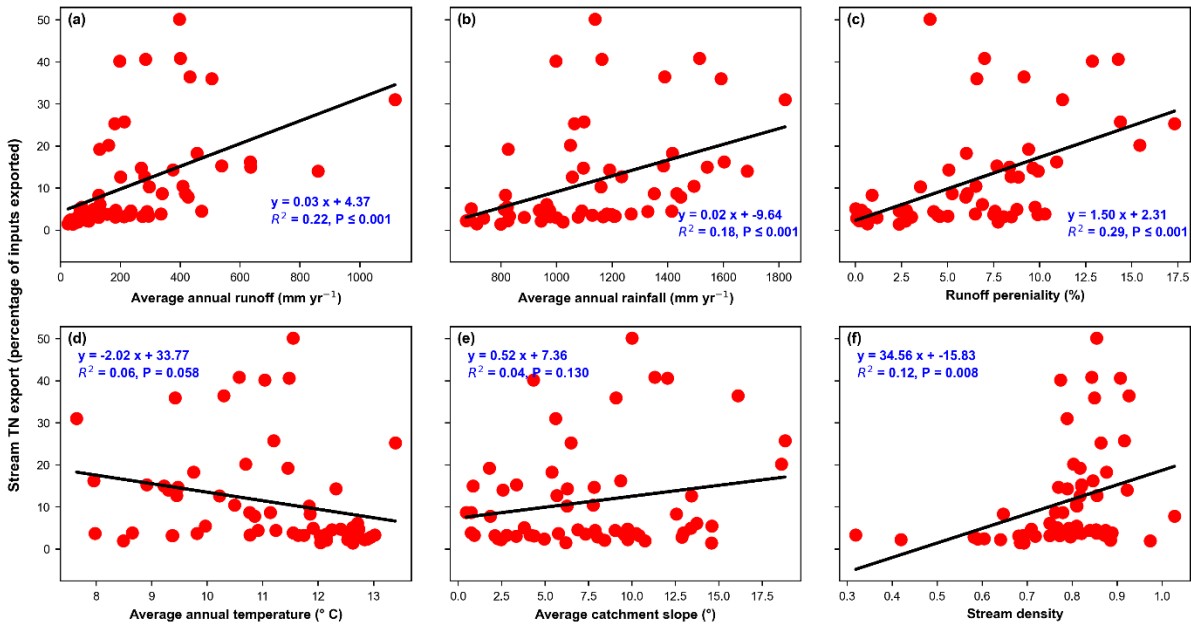

**Figure 9.** (a) Average annual streamflow, (b) Average annual rainfall, (c) Runoff perenniality, (d) Average annual temperature, (e) Catchment slope, and (f) Stream density vs. stream TN export (percentage of inputs exported).

## 4 Synthesis and Broader Implications

This section synthesizes our key findings, contextualizes them within the broader scientific literature, and discusses their implications for N management in diverse landscapes.

### 4.1 Variability in TN concentration and export

Our study reveals distinct N dynamics between forested and agricultural catchments. In forested catchments, despite the absence of fertiliser inputs, TN concentration (mean: 0.39 mg L$^{-1}$; range: 0.11–1.37 mg L$^{-1}$) and export (mean: 1.57 kg ha$^{-1}$ yr$^{-1}$; range: 0.32–7.36 kg ha$^{-1}$ yr$^{-1}$) exhibit substantial variability, likely driven by natural nitrogen cycling processes, microbial transformations, and hydrological factors. In contrast, agricultural land use has a pronounced impact on N dynamics. TN concentrations in agricultural catchments (mean: 1.57 mg L$^{-1}$; $p <$ 0.001) are nearly four times higher than in forested catchments, with stream nitrogen export 1.8 times greater from catchments typically with lower rainfall and runoff rates (mean 2.82 kg ha$^{-1}$ yr$^{-1}$; $p =$ 0.025). These differences highlight the dominant role of fertiliser application in elevating N levels and enhancing transport to streams in agricultural systems. Our findings align with global and Australian studies linking agricultural land use to elevated nutrient concentrations in water bodies (Djodjic et al., 2021; Liu et al., 2021, 2022a; Mitchell et al., 2009; Singh and Horne, 2020). These studies consistently identify agriculture as a key driver of N enrichment and spatial variability in N concentrations across streams. These results emphasize the critical role of land use in N transport and suggest that targeted management strategies are necessary for different land use types. While natural processes regulate N variability in forested landscapes, agricultural areas consistently exhibit elevated N concentrations and fluxes due to anthropogenic inputs.

**4.2 Export as a percentage of TN input: regional and global comparisons**

620 A key finding of this study is the relatively low percentage of TN export across the catchments, ranging from 1.43 % to 50 % of total nitrogen inputs, with a mean of 11 %. This is lower than estimates from other Australian catchments, such as the subtropical Richmond River Catchment (17 %, range 10–24 %; McKEE and Eyre, 2000), as well as global estimates, including northeastern U.S. watersheds (28 %, range 19–40 %; Schaefer and Alber, 2007), Lake Victoria (16 %; Zhou et al., 2014), and the Saint Lawrence Basin, Canada (22 %, range 11–68 %; 625 Goyette et al., 2016). Our estimates also fall well below the average reported for U.S. and European watersheds (25 %; Howarth et al., 2012). Our findings align with studies that have reported low percentages of TN export in other regions, such as southeastern U.S. watersheds (mean 9 %, range 5–12 %; Schaefer and Alber, 2007) and the Huai River Basin China (1.8 – 4.5 %; Zhang et al., 2015). They are also comparable to values reported for western U.S. watersheds (generally < 20 %; Schaefer et al., 2009). Notably, these studies, like ours, identified fertiliser as 630 the dominant source of N. In contrast, studies reporting higher percentages of TN export often identified other major N sources, such as atmospheric deposition (Boyer et al., 2002) or biological N fixation (McKEE and Eyre, 2000).

The relatively low TN export observed in our study suggests that a substantial portion of N is removed in produce, lost to the atmosphere, or retained within catchments, although its ultimate fate remains uncertain. 635 Because our export metric captures riverine export only (i.e., it does not account for nitrogen removed in produce), fertilised agricultural catchments may exhibit lower riverine export percentages simply because a sizeable share of N leaves the system via harvest. This raises important questions about whether fertiliser-based inputs inherently lead to lower export percentages, or whether retention and loss processes are more strongly governed by catchment-specific factors such as the extent of removal associated with agricultural production, hydrology, soil 640 properties, or microbial activity.

**4.3 Drivers of export as a percentage of TN input**

Our results demonstrate that hydrological factors, specifically runoff, precipitation, and runoff perenniality, are the primary drivers of the percentage of TN export across the studied catchments. Temperature, slope, and stream 645 density play secondary roles, suggesting that nitrogen export in these systems is predominantly driven by rapid hydrological flushing. This aligns with studies showing that short water residence times limit opportunities for biogeochemical processes such as denitrification, which requires prolonged interaction between N and microbial communities to significantly reduce N loads (Howarth et al., 2006; Seitzinger et al., 2006). These findings emphasize that regional differences in climate and hydrology drive variations in nitrogen export. In our wet 650 systems, where high precipitation and rapid runoff dominate, export is primarily controlled by hydrological flushing. By contrast, in the western U.S., stream export percentages were generally below 20% and were closely related to precipitation and runoff, reflecting hydrological limitation of N delivery in drier basins (Schaefer et al., 2009). In the southeastern U.S., export percentages were also low (5–12%) but were attributed primarily to warmer conditions enhancing in-system processing (e.g., denitrification) (Schaefer and Alber, 2007). This contrast shows 655 that low export percentages can arise from different mechanisms: in dry western basins, nitrogen delivery is limited by low precipitation and episodic runoff (Schaefer et al., 2009); in the humid southeastern U.S., similarly low export has been linked to stronger in-system processing under warm conditions (Schaefer and Alber, 2007).

While slope has been identified as a key driver of N export in some watersheds (Zhang et al., 2015), our findings suggest that in high-runoff environments, rapid water movement limits the influence of topography on N retention and export. Additionally, the moderate positive correlation between stream density and N export indicates that catchments with denser stream networks export a greater proportion of N inputs, potentially due to increased hydrological connectivity and reduced N retention time in soils. We note that export as a percentage of inputs is sensitive to input inventory uncertainty; see Section 4.5 for details.

## 4.4 Implications for catchment N modelling and management

The findings of this study provide valuable insights for advancing catchment N modelling and management strategies. The strong correlations between total and fertiliser N inputs and stream exports highlight the critical importance of accurate input estimation in predicting stream N loads. Our results emphasize the need for N models and management strategies that integrate both N inputs and hydrological processes. While N export is influenced by inputs, it is also strongly regulated by hydrology, retention mechanisms, and biogeochemical transformations. Furthermore, given that only a relatively small portion of N inputs is exported in streams, reducing inputs alone may not lead to a proportional decrease in this N export. This non-linear relationship should be considered in management strategies and policy development. A targeted approach, incorporating hydrological processes, N inputs, and catchment characteristics into modelling frameworks, can lead to more effective mitigation measures and policy interventions aimed at reducing N losses to waterways. This integrated approach will enable policymakers to make better-informed decisions that support sustainable catchment management practices. Ultimately, these strategies will help reduce nutrient loads, improve water resource sustainability, and protect aquatic ecosystems.

## 4.5 Limitations

During this study several limitations were identified, primarily related to data availability and quality, which may introduce uncertainty:

1. While rainfall correlated significantly with fertiliser nitrogen inputs; it did not fully explain variability at the CMA (regional) scale or capture farm to farm differences within regions. Dairy monitoring reports (Dairy Australia, 2021) show large differences in fertiliser use between farms within the same region. In addition, ABS aggregates nitrogen inputs into broad land use classes (grazing and cropping) and does not distinguish dairy, non-dairy livestock, mixed farming, or specific cropping systems. We addressed this using dairy and livestock monitor data, but mismatched data periods, especially for non-dairy livestock, may introduce temporal uncertainty and affect the accuracy of land use specific N input estimates.

2. Biological N fixation was estimated using simplified, land-use–specific literature values, which may introduce uncertainty in both magnitude and spatial variation.

3. Atmospheric deposition was not modelled due to the lack of spatially resolved data on wet and dry deposition, likely leading to an underestimation of total nitrogen inputs in some catchments.

4. Purchased feed and human food N were included using regional proxies. Although this approach expands the inventory, uncertainties persist in feed composition, stocking intensity, the proportion of metabolizable energy from purchased feed, and population estimates.

5. Errors in input estimates may bias export fractions. Underestimating total inputs (e.g., due to omitted deposition, small point sources, or uncertain fixation rates) would increase the calculated export fraction, whereas overestimating inputs would decrease it. Accordingly, we interpret export percentages comparatively and in the context of hydrologic controls rather than as precise mass-balance closure.

**5 Conclusions**

This study advances understanding of how N inputs, land use patterns, and hydrological processes collectively regulate N concentrations and export at the catchment scale. Our analysis revealed significant relationships between total N inputs, in-stream TN concentrations, and TN export. Land use emerges as a critical factor in N dynamics. While forested catchments generally exhibit better water quality due to the inverse relationship between forested land use and TN concentrations in surface water, TN concentrations and export remain variable. In contrast, agriculture, particularly dairy farming, is the dominant driver of N inputs and exports in Victoria. Although other high-input industries, such as certain horticultural operations, may contribute significantly at the farm level, their impact is not evident at the catchment scale used in this study due to their limited spatial granularity.

Our findings provide broader insights into N dynamics in regions with low fertiliser application rates and variable climates. In these landscapes, N export is primarily governed by hydrological flushing, with limited biogeochemical retention. This highlights the critical role of hydrology in N loss pathways and the importance of accounting for hydrological variability in N budgets and management strategies. These results have important implications for catchment management and policy development, emphasizing the need for integrated approaches that consider both N inputs and hydrological processes. Our study contributes to the global understanding of N dynamics by providing insights from a unique agricultural landscape with diverse climatic conditions. Future research should examine broader spatial scales to capture the full impact of land use on N dynamics and further explore the complex interactions between N inputs, land use, and hydrological processes in shaping N export patterns.

*Code and data availability.* Publicly available datasets were analysed in this study. The data sources are cited in the manuscript, and the processed data have been provided in the Supplementary Materials. Custom Python scripts used for basic data analysis are available from the corresponding author upon request.

*Supplement.* The supplementary material for this article is available in the online version at the publisher's website.

*Author contributions.* OB, MA, and AWW conceptualized the study and developed the methodologies. SNVN contributed to map creation. MA and AWW provided supervision. OB curated the data, wrote the original draft, and produced the figures, with feedback from MA, AWW, DG, SNVN, and IC. All authors reviewed and commented on the original draft and its revisions.

*Competing interests.* The authors declare that they have no conflict of interest

*Acknowledgements.* The authors wish to thank Perran Cook for his valuable discussions and insights.

*Financial support.* This research has been supported by the Australian Research Council via a Discovery Project (ARC, grant no. DP230100618). Olaleye Babatunde was supported by the University of Melbourne Research Scholarship.

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
