# Peer review of "Investigating Relationships Between Nitrogen Inputs and In-Stream Nitrogen Concentrations and Exports Across Catchments in Victoria, Australia"

_EGUsphere, 2025_

## Author Response (AR1)

**#Referee 1**

**We thank Referee 1 for their constructive review and thoughtful comments. Their feedback has been very helpful in improving the clarity and rigor of our work. Below we respond to each point in turn.**

**Reviewer comment**

I think it would be useful to either develop proxies for net food/feed inputs (say, based on even rough estimates of livestock numbers and population density as they vary by catchment) or, if this is not possible, simply frame the analysis as a relationship between rainfall-based estimates of fertiliser N loads and streamflow N, which is effectively what it is.

**Author Response**

We thank the reviewer for this valuable suggestion. In the revised manuscript, we have now included proxies for both livestock feed and human food nitrogen inputs. These additions are described in detail in Sections 2.5 and 2.6, with supporting data in Tables S7–S8 and Equations S1–S2. These additions expand the nitrogen input framework beyond fertiliser, aligning more closely with NANI approach.

**Reviewer comment**

Given the seemingly strong relationship between elevation and rainfall shown in Figure 1, it is surprising that elevation is nowhere mentioned as an explanatory variable, nor included in any tables. Was it investigated?

**Author Response**

Thank you. We did not include elevation as a predictor. Our modelling uses rainfall as the hydrometeorological driver most directly linked to moisture availability and agricultural intensity in Victoria. Because elevation closely covaries with rainfall regionally, adding it would risk collinearity without providing independent explanatory value.

**Reviewer comment**

Line 183: Need to cite a reference for QGIS.

**Author Response**

We have added the official citation for QGIS as recommended. Section 2.3 now reads… using QGIS (QGIS Development Team, 2025).

**Reviewer Comment**

Adams et al. (2014) considered only wet deposition of N and did not distinguish between reduced and oxidised forms. Reduced N (including $NH_3$, $NH_4^+$, etc.) are well known to be associated with volatile losses of $NH_3$ from agricultural sources including manure from dairy herds and fertiliser applications, which are likely relevant here. … Either do a better job of estimating the true atmospheric deposition component and its spatial variation or drop it as insignificant.

**Author Response**

We appreciate this valuable comment. In the revised manuscript, we excluded atmospheric N deposition from the catchment input calculations because spatially explicit data distinguishing reduced and oxidised N forms were unavailable for Victoria. The description in Section 2.7 was updated accordingly, and this omission is now explicitly acknowledged as a limitation in Section 4.5 (point 3).

**Reviewer Comment**

What is "fertiliser additive land use"?

**Author Response**

Thank you for pointing this out. The term has been corrected to "agricultural land use percentage", defined as the share of catchment area under agricultural uses.

**Reviewer Comment**

Schaefer et al. (2009) actually show lower export as a percent of N inputs in some western U.S. watersheds than Schaefer and Alber (2007) show in southeastern U.S. watersheds, presumably because of the relatively dry environments in much of the west which affect N delivery (as you note in Section 4.3).

**Author Response**

Thank you for this clarification. We revised Sections 4.2 and 4.3 to correctly describe that Schaefer et al. (2009) reported lower export in western U.S. watersheds due to hydrologic limitation in drier climates, whereas Schaefer and Alber (2007) observed similarly low export in the humid southeast attributed to enhanced in-system processing under warm conditions.

**Reviewer comment:**

A potential difficulty with the discussion of TN export as a fraction of inputs is that the estimated total catchment N inputs may be biased because of incomplete estimates. Some discussion of this specific issue is warranted.

**Author Response**

We agree with this observation and have added an explicit statement acknowledging that uncertainty or bias in input estimates can affect calculated export fractions. This discussion is now included in Section 4.5 (Limitation #5) and referenced at the end of Section 4.3

**#Referee 2**

**We thank Referee 2 for their constructive comments, which have been very helpful in improving the clarity and rigor of our work. Below we respond to each point in turn.**

**Reviewer comment:**

Firstly, the fertilizer model relies solely on mean annual rainfall as a predictor for four distinct agricultural land uses (presumably dairy, cropping, etc.). However, irrigation is also very important in this arid zone. How much the N input by irrigation? And did irrigation rates could affect this model?

**Author Response**

We appreciate this important point. Our regression was fitted using mean annual rainfall (MAR) at the CMA-by-land-use scale, but irrigation effects are explicitly incorporated at the parcel scale when mapping estimates. In the revised Section 2.3, we clarify that parcels mapped as irrigated use total water input (TWI = MAR + mean irrigation depth), whereas non-irrigated parcels use MAR. This adjustment applies to 7.8% of the mapped agricultural area in the Victorian land-use data.

Regarding "N input by irrigation": our study estimates fertiliser N applied to land and does not quantify nitrogen delivered in irrigation water. That flux is outside the scope of this analysis. Accordingly, the irrigation term serves as a water-supply modifier that captures irrigation's indirect effect on fertiliser demand via TWI, rather than a direct N addition.

On whether irrigation rates affect the model: irrigation does not affect the fitted coefficients (which are MAR-based), but it does influence parcel-level predictions in irrigated areas through the TWI substitution (Greater irrigation depth increases total water input (TWI), which in turn increases the estimated fertiliser N for otherwise similar parcels). We have made this workflow explicit in Section 2.3 and report the statewide prevalence of irrigated area for context.

**Reviewer comment:**

Line 53-54 The phrase "complex to manage and modify" is redundant and awkwardly implies that modifying the models (rather than their operation) is the focus.

**Author Response**

We thank the reviewer for this clarification. Our intention was to emphasise application challenges due to data demands, not model modification. The revised manuscript now reads:

"However, these process-based models are challenging to apply because of their extensive input requirements (e.g., climate variables, soil properties, and agricultural management such as crop rotations, fertiliser application, and irrigation)."

**Reviewer comment:**

Line 57: The comparison is syntactically incomplete. "Unlike process-based models" lacks a clear verb to contrast what those models do differently and Line 58: Absence of 'that' before 'N undergoes' creates a grammatical error.

**Author Response**

We agree that the sentence was syntactically incomplete. The revised manuscript now reads:

"Unlike process-based models, which simulate the internal cycling and transformation of nitrogen within the catchment, the NANI approach compares nitrogen inputs to exports without accounting for these internal processes".

**Reviewer comment:**

Line 126: The phrase "geographic representation across..." is slightly awkward. The concepts of "data availability" and "representation" are not perfectly parallel. The intent is clear but can be expressed more forcefully and directly.

**Author Response**

We agree. The revised manuscript now reads: "These sites were selected to ensure long-term data availability and broad geographic coverage of diverse land uses and climatic conditions across the state."

**Reviewer comment:**

Line 325: The pronoun "That" is slightly vague. While it logically refers to the data collection period of Gourley et al. (2008-2009), the reference can be made more explicit for immediate clarity.

**Author Response**

We agree. The revised manuscript now reads: "The 2008–2009 data collection period overlapped with the Millennium Drought, which likely reduced fertiliser application rates and may explain why the median reported by Gourley et al. (2012) is lower than the range observed in this study."

**Reviewer comment:**

Line 347: 'natural soil N' is vague. In soil science, the preferred terms are typically 'native soil nitrogen' or 'soil nitrogen mineralization.'"

**Author Response**

We agree. The revised manuscript now reads: "…These farms often rely on alternative N sources, such as native soil nitrogen and biological N fixation by leguminous crops, to meet their N requirements (Angus, 2001)."

**Reviewer comment:**

Line 364: The phrase "cropping land use, including horticulture" is problematic. In standard agricultural classification, "horticulture" (intensive, high-value crops like vegetables and fruits) is often considered distinct from broadacre "cropping" (extensive field crops). Grouping them under one "cropping land use" umbrella is confusing and requires justification, as their nutrient management practices differ vastly.

**Author Response**

Thank you for flagging this. We clarified the terminology and the data constraint. Our inventory source reports fertiliser inputs in a single "cropping" class that aggregates broadacre and horticulture at the mapping scale used here; separate horticultural inputs were not available. Because horticulture represents a small share of the total cropping area in our region, we retained the combined category and noted this as a data limitation. This has been captured in the revised manuscript (Section 3.2.3).

**Reviewer comment:**

Line 380: While "data" is technically plural, it is often treated as a singular mass noun in scientific writing, especially when referring to a dataset as a whole. Using the singular verb is more modern and common.

**Author Response:**

We agree. The revised manuscript now reads: "The data from Fig. 4 was aggregated to calculate the average TN input across each catchment…"

**Reviewer comment:**

Line 451: The space between the numeral and the percent symbol ("43.4 %") is a typesetting style often required by publishers. However, many style guides (e.g., APA) recommend no space ("43.4%"). The key is to be consistent throughout the manuscript.

**Author Response**

Thank you. We follow BG/Copernicus style, which uses a space between the numeral and the percent sign. The manuscript has been standardised accordingly (e.g., "43.4 %").

**Reviewer comment:**

Line 597: The verb "indicate" is weak and overused in scientific writing. A more assertive and precise verb would strengthen the opening statemen

**Author Response**

We agree. The revised manuscript now reads. "Our results demonstrate that hydrological factors, specifically runoff, precipitation, and runoff perenniality, are the primary drivers of the percentage of TN export across the studied catchments."

**Associate Editor comment:**

A major revision is needed to further clarify the models (e.g., the impacts of net food/feed imports and irrigation on N transport).

**Author Response**

We thank the Associate Editor for this valuable feedback and have substantially revised the manuscript to address these key points.

- Net food and feed inputs: We have now included proxies for both livestock feed and human food nitrogen inputs, as recommended. These are described in Sections 2.5 and 2.6, with supporting data and equations provided in the Supplement (Tables S7–S8 and Eqs. S1–S2). Their inclusion strengthens the representation of non-fertiliser nitrogen sources within the overall nitrogen budget.
- Irrigation effects: Section 2.3 has been expanded to clarify how irrigation is incorporated in the modelling workflow. Specifically, irrigation depth is added to mean annual rainfall (MAR) to derive total water input (TWI = MAR + irrigation depth) for irrigated parcels, representing irrigation's indirect influence on fertiliser demand and its contribution to nitrogen inputs in irrigated regions.